# XML-Based Automatic NIOS II Multi-Processor System Generation for Intel FPGAs

**Haotian Cao ***  and Uwe Meyer-Baese

Department of Electrical and Computer Engineering, FAMU-FSU College of Engineering, Tallahassee, FL 32310, USA
* Correspondence: hc18f@fsu.edu

**Abstract:** Many embedded systems are introducing processing units to accelerate the processing speed of tasks, such as for multi-media applications. The units are mostly customized designs. Another method of designing multi-unit systems is using pre-defined standard intellectual properties. However, the procedure of arranging IP cores in a system and maintaining a high performance as well are the remaining challenges. Implementing softcore processors on field-programmable gate arrays (FPGAs) is a relatively fast and inexpensive choice to design and validate a desired system. This paper describes the rapid prototyping of hardware/software co-design based on FPGAs. A novel system generator to effortlessly design a multiple NIOS II soft-processor core systems is also purposed. The NIOS II CPU is a configurable RISC processor designed by Altera/Intel and can be trimmed to complete specific tasks. The error-prone and time-consuming process of designing an IP block-based system is improved by the new novel system generator. The detail of the implementation of such system is discussed. To test the performance of a multi-NIOS II system, a parallel application is executed on 1-, 2-, 5-, and 10-core NIOS II systems separately. Test results prove the feasibility of the proposed methodology (for an FIR filter, a dual-core system is 29% faster than a single-core system; a 5-core system is 28% faster than the dual-core system).

**Keywords:** FPGA; soft-core processor; rapid prototyping; multi-core system

## 1. Introduction

The requirement of high computational ability for multi-media applications catalyzes the uses of specific processing units in embedded systems. Many embedded systems consist of a fixed processor and a series of peripherals. These traditional embedded systems have filled in the needs of proceeding with data-intensive computations. However, the lengthy design period and high price have caused their low adoption rate [1]. Due to the fast-changing market, designing a specific system is a risky and costly undertaking for a company. To design a configurable system with lower cost, field-programmable gate arrays (FPGAs) have been increasingly adopted as a design target. Their flexible architecture and parametric property have fulfilled the variant requirements from developers. Plentiful on-chip logic elements and high-bandwidth I/O have been applied on many image processing applications. For instance, Ma et al. [2] proposed a tool for porting Deep Neural Network to FPGA with a throughput of over 700 giga operations per second. Developers have also proved the benefit of its flexibility. Garcia et al. [3] connects a frame buffer to memory resources of an FPGA so as to complete the task of random access for the whole image frame, which is difficult to directly realize on the memory on FPGAs. Standard intellectual property (IP) cores are supported by FPGAs, which can be easily reused and can complete the mapping of different logic resources quickly. Similarly, the design of soft processor cores uses programmable logic resources on FPGAs to realize the logic of a processor. The two FPGA market leaders provide two commercial soft cores come from Xilinx (MicroBlaze) [4] and Altera/Intel (NIOS II) [5]. Much research has been developed by researchers recently

around soft processor cores. For instance, Siddiqui et al. [6] proposed a 16-bit processor (IPPro), which operates at 337 MHz to execute tasks for image processing algorithms. Ammar et al. [7] demonstrated a DSP processor on Xilinx Virtex 5, which can enhance the existing applications (FIR filter and Matrix Multiplication) by 900% on average.

In recent years, a lot of research has embedded soft processor cores into its system designs. Danial et al. [8] proposed a miniature, battery-operated FPGA platform for wearable and IoT computing. The NIOS II processor was used to perform the FIR algorithm on the standalone design. A hybrid energy storage system based on FPGA was described by Hatim et al. [9]. The validation of its energy management program using fuzzy logic controller was executed on an NIOS II processor. Vilabha et al. [10] demonstrated a soft-core-based wireless sensor network node. The NIOS II processor is used for a single node to accelerate complex algorithms. The use of soft-core processors has significantly enhanced the performance of clock cycle counts. Arun et al. [11] realized a NIOS II-based system for a power flow management control algorithm. The system was running on Cyclone-III and was tested with a real-time scenario. In the scenario, solar photovoltaics were the energy source, and a lead–acid battery was the storage.

The flexibility of FPGAs also features it as a feasible platform for parallel systems. A multi-processor system on an FPGA can satisfy the demands of power, area, and cost and facilitates the designing progress. Eric et al. [12] presented a modified multi-MicroBlaze soft-core platform with Linux Symmetric Multi-processor support. Furthermore, they discussed the performance of this processor after providing the L1 data cache support to this platform [13]. Daniel et al. [14] proposed a novel Multi-Processor System-on-a-Chip (MPSoC), which provides multiple segregated subsystems with a shared memory. It eliminated the drawback of the software partitioning for the segregated system since each subsystem is a separate SoC.

By utilizing the IDE toolchain provided by FPGA manufactures, researchers can acquire a desired multi-processor solution. However, the development of hardware/software (HW/SW) co-design using the toolchain and the construction of an IP block-based system are error-prone and time-consuming. This article purposes a novel multi-CPU system generator which can effortlessly build a multi-NIOS II system that does not require researchers to set up any configuration to utilize.

The remainder of the paper is presented as follows. Section 2 describes background knowledge about FPGAs, the NIOS II processor, HW/SW co-design with Quartus toolchain, and memory requirements for running an NIOS II processor on FPGAs. Section 3 provides the mechanism of the multi-CPU system generator and demonstrates an FIR filter as a parallel application running on multi-core systems. Section 4 provides the performance result of the FIR filter. Section 5 discusses the design approaches. Section 6 concludes the article.

## 2. Background

### 2.1. Field-Programmable Gate Arrays

#### 2.1.1. Overview of FPGAs

A field-programmable gate array is a two-dimensional programmable device that consists of logic cells and programmable switches placed in columns and rows. A simplified diagram of the FPGA structure is shown in Figure 1. A generic logic cell can execute a single task, and interconnections of the logic cells are established by programmable switches in between the logic cells. By using hardware description language, developers can assign functions to each logic cell and trigger the programmable switches to transmit data and finally acquire a customized system.

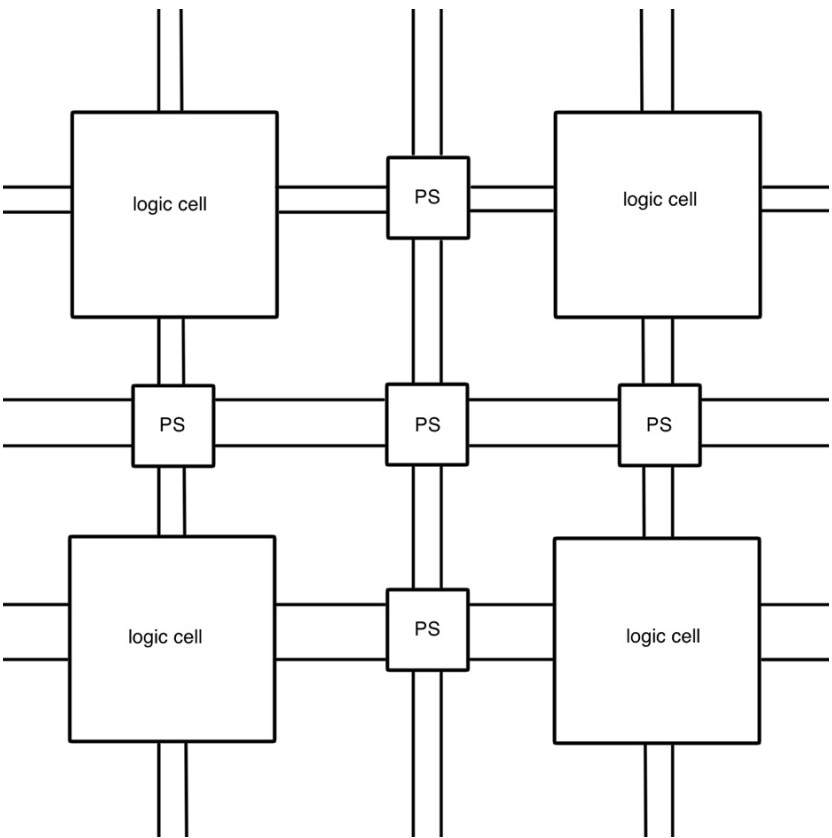

**Figure 1.** A simplified diagram of FPGA structure.

A logic cell consists of a lookup table (LUT) and a D flip-flop (D FF). It uses the lookup table as a combinational circuit to execute any combinational function. FPGA manufacturers choose different numbers of input of LUT for their FPGAs. As shown in Figure 2, a three-input LUT-based logic cell has three inputs (a, b, and c) with a D FF block. FPGA can form a sequential circuit or a combinational circuit by choosing one of the outputs of the logic cell, that is, y as combinational output and q as sequential output. Macroblocks are integrated in many FPGAs. These cells are responsible for operating tasks other than computation such as memory blocks, clock management, and I/O interfaces.

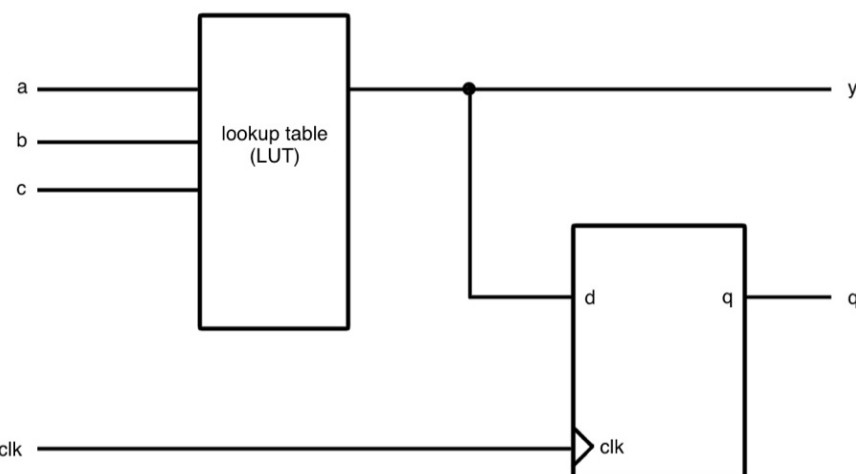

**Figure 2.** A diagram of the three-input LUT.

2.1.2. The Structure of Stratix IV GX

The 40 nm programmable logic devices named as Stratix IV were introduced by Altera/Intel in May 2008. These devices include Stratix IV GT, GX, and E series. The features of three variants are displayed in Table 1. The GT and GX series have similar capability, while the GT series has a higher transceiver performance. E (enhanced devices) series focus on a large amount of logic resources (LEs and RAM size) without any transceiver integrated in.

**Table 1.** Features of Stratix IV series.

|  | **Stratix IV GT** | **Stratix IV GX** | **Stratix IV E** |
|---|---|---|---|
| Transceivers | up to 11.3 Gbps | up to 8.5 Gbps | None |
| Logic elements | up to 530 K LEs | up to 530 K LEs | up to 820 K LEs |
| RAM size | up to 27,376 Kbits | up to 27,276 Kbits | up to 33,294 Kbits |

A chain of the generic logic cells is called a Logic Array Block (LAB) that can perform arithmetic functions (addition and subtraction), logic functions (and, or, etc.), and register functions (storing values). A structure of LABs and their interconnects are presented in Figure 3. LABs and Macro Cells can deliver signals between column and row interconnects [15]. C4/C12 represents a length of 4 or 12 blocks to the right or the left based on the column channel. R4/R20 represents a length of 4 or 20 blocks up or the down based on the row channel. Signals can be sent to the LAB directly by local interconnect. The neighbor LABs can be connected by direct link interconnect. Note that ALM (adaptive logic module) is a variant of the logic cell. An ALM's output can be connected to any type of interconnects based on the design.

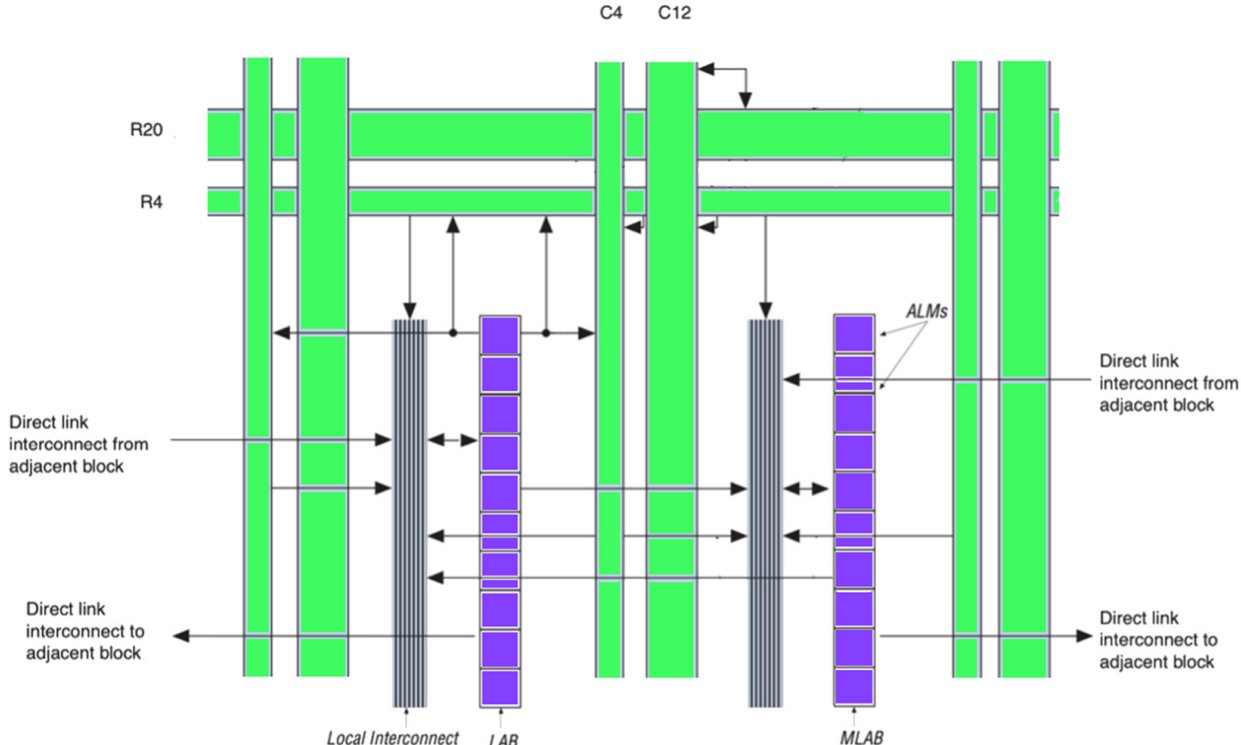

**Figure 3.** Stratix IV LAB structure and interconnects.

The design of this article is implemented on both the Terasic DE-4 (Stratix IV GX EP4SGX230C2) [16] and DE-1 (Cyclone V 5CSEMA5F31C6) [17] FPGA board. Table 2 shows the logic resources of the two FPGAs. It can be observed that DE-4 is a device

that has a larger amount of logic resources than DE-1. Especially, 17,133 Kbits of on-chip memory can store sizable input data and software designs compared with 4450 Kbits on the DE-1 board. Multiplier blocks can be utilized for DSP computation. To synchronize the clock phase between on-chip and off-chip devices, phase-locked loops (PLLs) are required in order to prevent delays, such as integrating an off-chip memory device into a customized system.

**Table 2.** Logic resources of the DE-4 and DE-1 FPGA boards.

| | Logic Elements | On-Chip Memory | Multiplier Blocks | PLLs | Off-Chip Memory |
|---|---|---|---|---|---|
| DE-4 | 228,000 LEs | 17,133 Kbits | 1288 18 × 18-bits blocks | 8 PLLs | 64 MB Flash; 2 MB SSRAM; 2 Kbits EEPROM |
| DE-1 | 85,000 LEs | 4450 Kbits | None | 6 PLLs | 64 MB SDRAM; EPCS128 |

*2.2. NIOS II Soft-Processor Core*

2.2.1. Overview of NIOS II Processor

The NIOS II is a RISC (reduced instruction set computer)-based configurable soft-core processor. Unlike traditional prefabricated hard-core processors, it is described by synthesizable, encrypted HDL codes that implement the functions of a processor with the mapping of generic logic resources. Developers can trim the processor to meet the specific requirements. It retains the characteristics of the RISC architecture:

- 32-bit instruction, data path, and address space;
- Two types of instructions: memory access and ALU operations;
- Same address space that represents both memory and I/O devices;
- 32-level IRQ (interrupt) requests.

To satisfy various needs of an embedded system, Altera/Intel releases three versions of NIOS II [18]:

- NIOS II/f: The fastest version is aimed for peak performance. It has a 6-stage processor pipeline, dynamic branch prediction, and optional instruction and data cache;
- NIOS II/s: The standard version has a 5-stage processor pipeline, static branch prediction, and optional instruction cache;
- NIOS II/e: The slowest version is aimed for smaller sizes. There is no cache included in it and is not pipelined.

Despite the modification that has been accomplished by Altera/Intel, users can configure the processors to adjust the size, functions, and performance. For instance, the data and instruction cache can be included in NIOS/f. The size of two caches can be tuned from 0.5 KB to 64 KB. Moreover, unnecessary units can be removed to reduce the size of the circuit, such as the JTAG debug module or a custom instruction logic unit. Table 3 demonstrates the performance of each version.

**Table 3.** Performances of each NIOS II version.

| | NIOS II/e | NIOS II/s | NIOS II/f |
|---|---|---|---|
| Number of Logic elements | 540 Les | 1030 Les | 1600 Les |
| Max clock frequency | 195 MHz | 110 MHz | 140 MHz |
| Performance | 18 MIPS | 55 MIPS | 145 MIPS |

2.2.2. Mechanism of Memory Allocation

For a small-sized design, on-chip memory can be the only device that stores instructions and data of NIOS II processors. Figure 4 shows an NIOS II processor program assigned in memory space. The program starts from 0 × 0000 to 0 × 1FFF [19]. The NIOS II

software build tool (SBT) determines the starting point by a processor's exceptional address. Without any specific requirement, the end of the program is the end of the memory space. It has five sections. The NIOS II SBT links the program to memory space. Therefore, it is also named as linker sections. Usually, the five sections are connected in sequence, but they can be allocated to nonadjacent memory space by specification. The default sections are listed below:

- text: The executable code of the program;
- rodata: Read-only data defined in the .text section;
- rwdata: Variables and pointers stored in this section. Values can be read and written;
- heap: Dynamically allocated memory address is located;
- stack: Temporary data and function parameters are stored in this section.

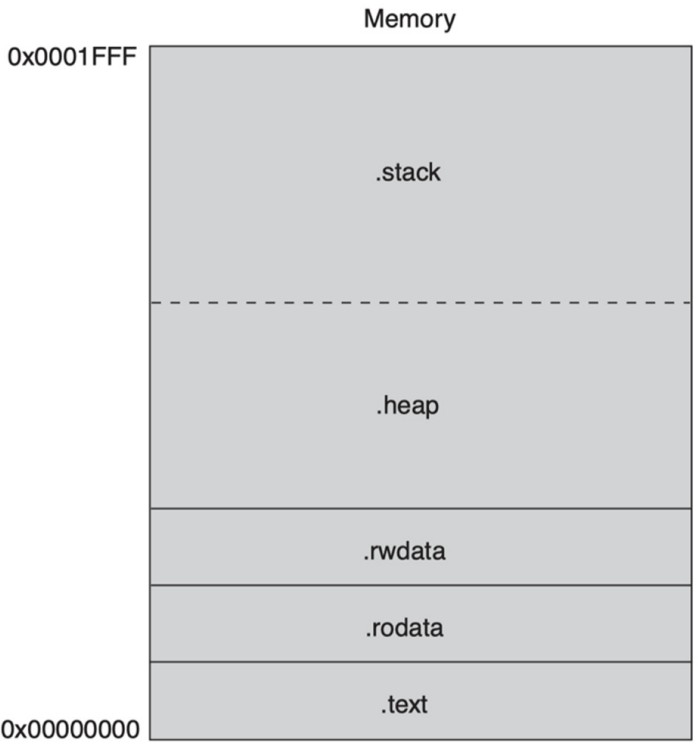

**Figure 4.** An NIOS II program in on-chip memory (on-chip RAM).

For a large design, a more complicated booting process is required. First, an off-chip RAM and an off-chip ROM are needed in the system. The NIOS II SBT will wrap the program up with a boot loader (or boot copier) when the program is downloaded to the ROM. When the program starts to run, the boot loader will copy the entire program data to the RAM. Then, the program will execute similarly as in Figure 4. Figure 5 demonstrates an NIOS II program stored in an off-chip ROM.

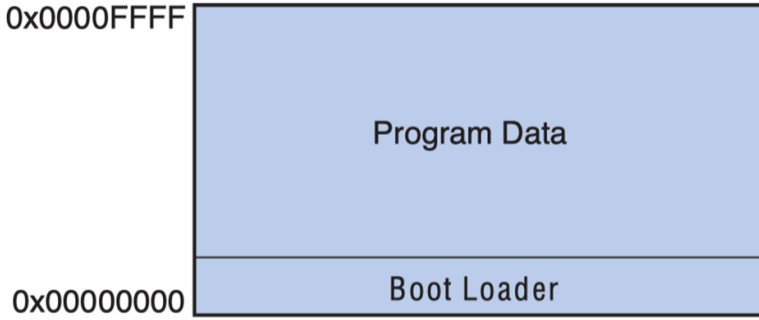

**Figure 5.** An NIOS II program in on-chip memory.

Table 4 lists some possible NIOS II boot methods provided by Altera/Intel [20]. The process of booting on CFI and UFM are similar. The boot copiers are placed in CFI flash or UFM themselves. They are pointed by reset vectors. EPCS/EPCQ flash does not store the boot copier. Instead, it is stored in a part of the on-chip RAM that is utilized by the EPCS/EPCQ controller. When the program starts to run, it looks for the reset address which is pointing to the EPCS/EPCQ controller. Then, the boot copier executes and copies the program from EPCS/EPCQ flash to a RAM device which is pointed by the exception vector.

**Table 4.** Generic boot method of NIOS II program.

| Boot Option | Reset Vector Configuration | Exception Vector Configuration |
|---|---|---|
| Boot copier in CFI flash | CFI flash | On-chip or off-chip RAM |
| Boot copier in EPCS/EPCQ flash | EPCS/EPCQ controller | On-chip or off-chip RAM |
| Boot copier in UFM | UFM | On-chip or off-chip RAM |
| Directly boot in on-chip RAM | On-chip RAM | On-chip RAM |

### 2.2.3. Hardware Abstraction Layer Paradigm

The application was executed by the hardware abstraction layer (HAL) paradigm. The HAL paradigm is a solution in between a barebone system and a full function OS. It consists of three sections: C library, HAL application interface (API), and device drivers. Figure 6 displays the structure of the software hierarchy. Intel/Altera provides device drivers for commonly used IP cores and libraries for accessing the custom hardware. Drivers and libraries are the connections between HAL API and the custom hardware. HAL API includes functions to invoke hardware. Therefore, developers can directly write and test a C-language-based program without installing any system. Note that the HAL paradigm needs to be enabled in IDE during software development.

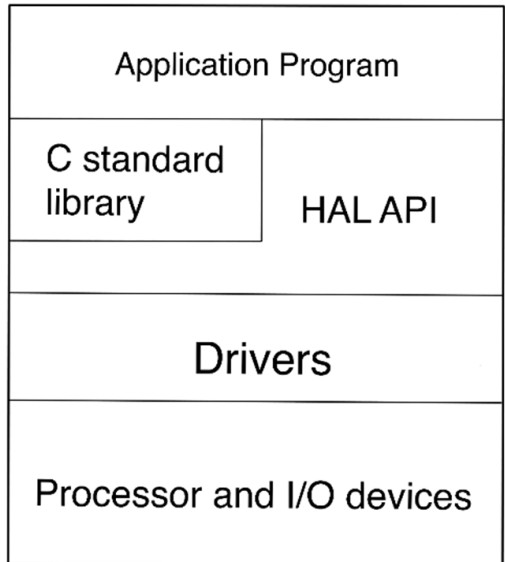

**Figure 6.** The structure of the HAL software hierarchy.

### 2.3. Required/Used Property (IP) Cores in Multi-uP Systems

### 2.3.1. On-Chip Memory Core

Compared with off-chip memory devices, on-chip memory blocks have a smaller size and faster speed. Common on-chip memory blocks on the Stratix IV family are listed in Table 5 [21]. The total size of M144K and M9K includes parity bits. The default size is less than the total size. For instance, M144K blocks have 147,456 (144 K) bits in total. Only 131,072 (128 K) bits are utilized by default. Each type of memory block has different access speeds.

**Table 5.** On-Chip memory blocks on Stratix IV family.

| Feature | M144K | M9K | MLABs |
| --- | --- | --- | --- |
| Total size per block | 144 Kbits | 9 Kbits | 640 bits |
| Default configurations (depth $\times$ width) | 128 K $\times$ 1 | 8 K $\times$ 1 | 64 K $\times$ 10 |
| Maximum performance | 540 MHz | 600 MHz | 600 MHz |

Memory blocks on FPGAs can be instantiated by calling an on-chip memory (RAM and ROM) core. Developers can change the configurations to customize a memory device in the system. Important configurations are listed below:

- Block type: Can only choose a single type of the memory blocks for one memory IP core;
- Data width: Can choose from 8 to 1024 depending on the design;
- Total memory size: Can choose the size that does not exceed the maximum size of a memory device;
- Memory initialization: Can enable or disable the initialization of the memory content. Developers can also download input data when the memory is initialized.

### 2.3.2. Mutex Core

To prevent the overlapped access of the memory address in a multi-processor system, a mutex core is required in the system. It is a protocol to provide mutual exclusive access of a shared device. When a processor needs to access a shared device, it will access the register of the mutex core to ensure if the device is occupied. If it is not occupied, the processor will claim the ownership of the mutex by writing its CPU ID to the OWNER section of the mutex register and changing the VALUE section to a non-zero value. Once the access of the device is completed, the processor will release the mutex core by clearing the corresponding register. A processor will constantly check the register when the core is owned by another processor until the core is released. The reset register will need to be set high to enable the core after the system reset. Figure 7 displays the register map of a mutex core [22].

| Offset | Register name | R/W | Bit description | | |
| --- | --- | --- | --- | --- | --- |
| | | | 31-16 | 15-1 | 0 |
| 0 | mutex | RW | OWNER | VALUE | |
| 1 | reset | RW | Reserved | | RESET |

**Figure 7.** Register map of a mutex core.

### 2.3.3. JTAG UART Core

The communication between an FPGA and a host PC is established by a JTAG controller integrated in the FPGA. A download cable connects the FPGA and the host to transmit data streams. On the host PC side, developers use the IDE or JTAG terminal to input commands and display output messages. On the FPGA side, the NIOS II JTAG debug module and JTAG UART module are connected. The JTAG debug module can be used for adjusting the NIOS II processor. The JTAG UART module is used for transferring the output messages or data. The JTAG UART IP core instantiates the JTAG UART module. In a multi-processor system, multiple JTAG cores can be called. However, only one core can access the download interface at a time. Figure 8 shows a conceptual diagram of the communication between an FPGA and a host PC.

### 2.3.4. Interval Timer Core

An interval timer can be assigned by its IP core. In NIOS II SBT, a timer can be set as a system timer or a user timer. A system timer runs continuously for interrupted service in the system. A user timer can be used as a snapshot timer. Developers use this timer to

record the current clock cycles in the system. The system timer and the user timer should not be the same timer. In a multi-processor system, multiple timer cores can be included.

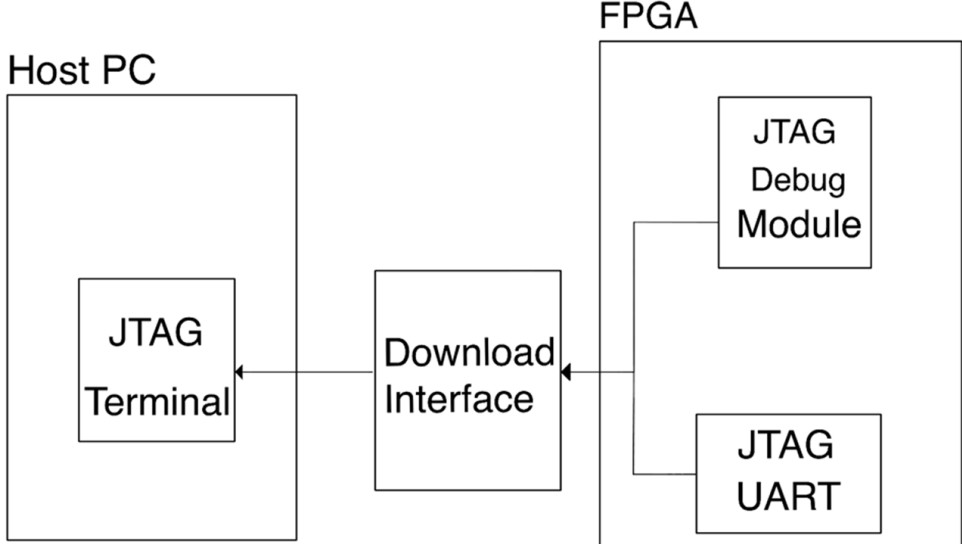

**Figure 8.** A conceptual diagram of the communication between an FPGA and a host.

## 3. Methodology

### 3.1. The Process of HW/SW Co-Design

Figure 9 describes the common process of developing an embedded system on the Altera/Intel FPGAs. At first, developers specify and separate the functions of the hardware and software. On the hardware developing part, a hardware design will be built on platform designer software. It is saved as a .qsys file. After generating a .qsys file, a <.sopcinfo> file will be created and will be read by a board support package (BSP) generator to link the software design and the hardware. On Quartus, a <.verilog> file as a top-level design is required to instantiate the design of the platform designer. An FPGA image (<.sof> file) will be generated after the compilation on Quartus. Afterwards, it can be programmed into the FPGA. On the software developing part, a conceptual design which includes the functions of the application can be developed before the hardware development. However, a complete finished C code can be developed only after the <.sopcinfo> file is generated by BSP generator. In the BSP generator, developers can optionally choose to use a hardware abstraction layer (HAL) library or a full-featured C library. The HAL library facilitates the utilizations of the IP core. By calling HAL functions of the IP core, developers do not need to access the specific register to use the specific feature. Full-featured C library supports floating points computation and other popular functions. A small-sized C library lacks the support of many functions but occupies less memory space. On Stratix IV, the size of a Hello World example with a full-featured C library is 70 KB. On the contrary, with a small-sized library, the size of an .elf file is 2 KB.

### 3.2. The Mechanism of Multi-processor Generators

#### 3.2.1. The Design of a Multi-NIOS II System

The challenges of building a multi-NIOS II system can be addressed as three parts:

- The application memory space of each processor;
- Booting methods for a multi-NIOS II system;
- The protection of the access of a shared device.

Figure 10 shows the memory space of a 3-core multi-NIOS II application. For a large application, the application is usually downloaded to an off-chip ROM. Unlike the single-core application, the end of the previous application is the start of the next application. Boot loaders will copy applications to RAM when the program starts. In this case, the

reset vector of each processor is set to the ROM, and the exception vector is set to the start address of the application. Developers can adjust the space of each application by changing the vectors in the system. If the application size is smaller than the maximum size of the on-chip RAM, it is capable of directly booting a multi-core system in on-chip RAM by setting both reset and exception vectors to on-chip RAM.

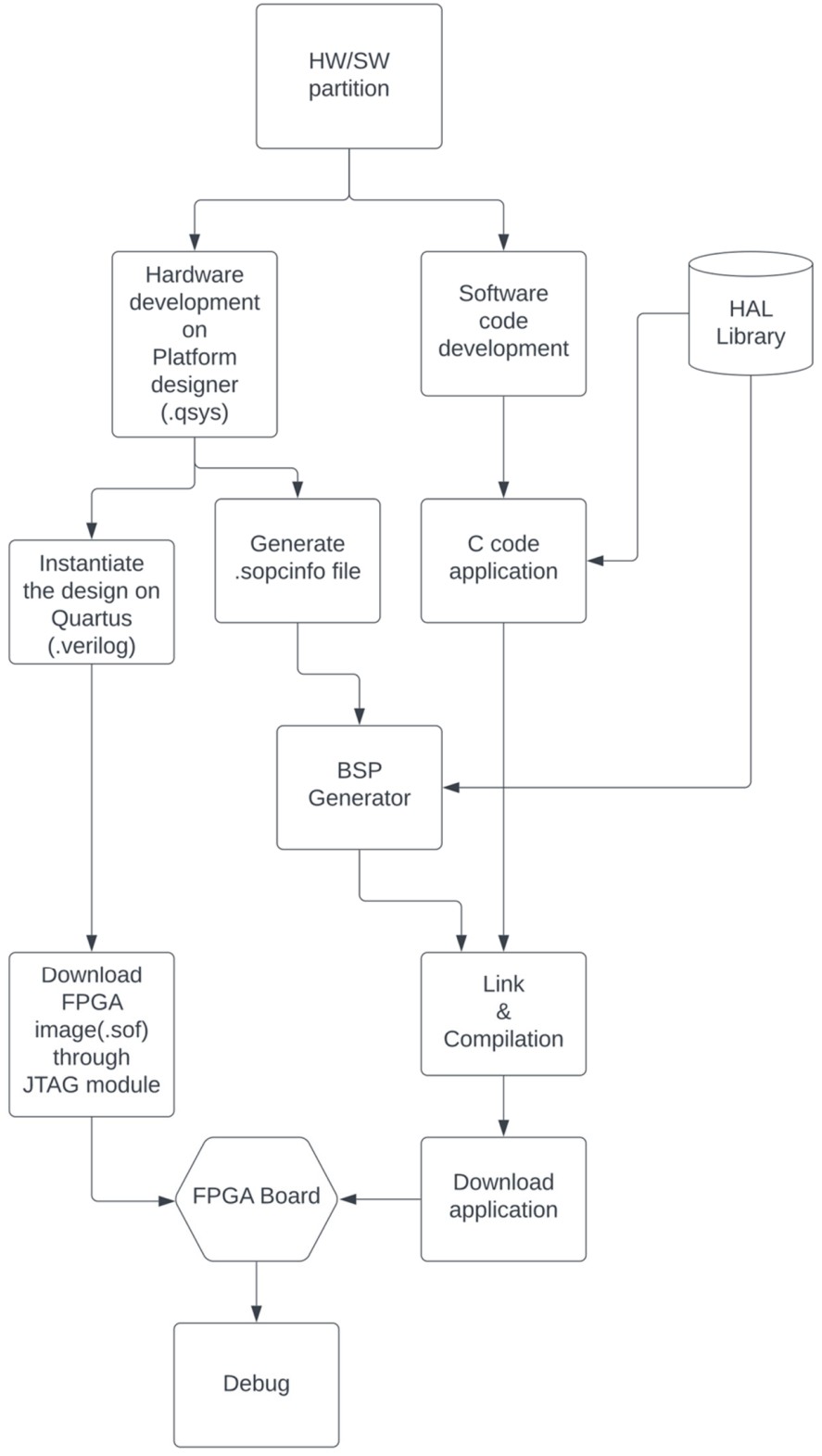

**Figure 9.** Common process of HW/SW co-design.

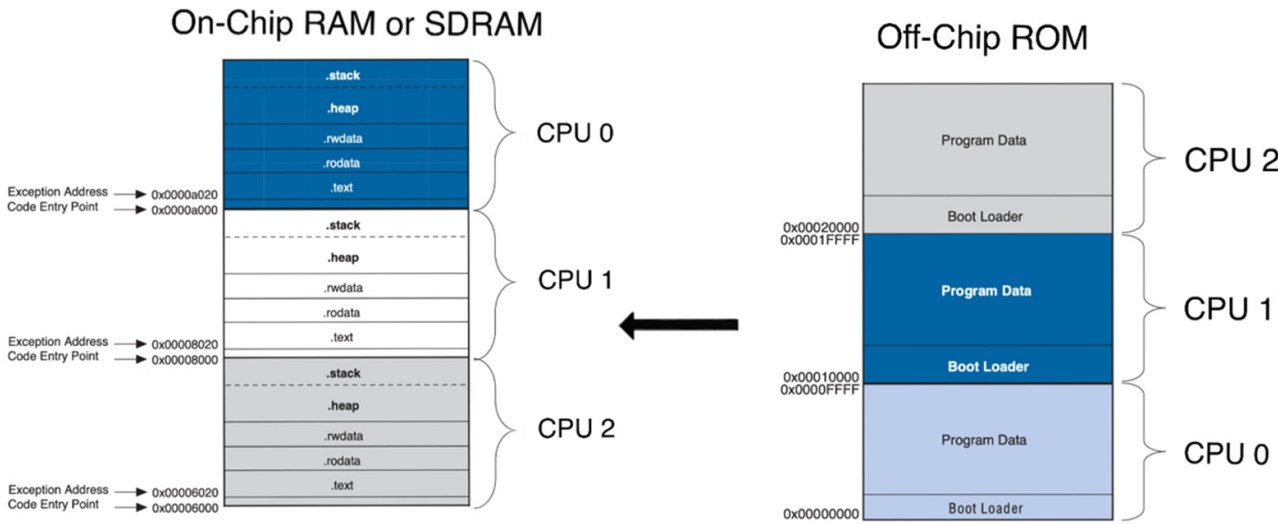

**Figure 10.** The memory space of a 3-core multi-NIOS II application.

Figure 11 demonstrates a diagram of a multi-NIOS system with peripherals. Each CPU connects to a system ID core, a JTAG core to communicate with the host PC, and a timer core. The accesses of the shared on-chip memory are protected by a mutex core.

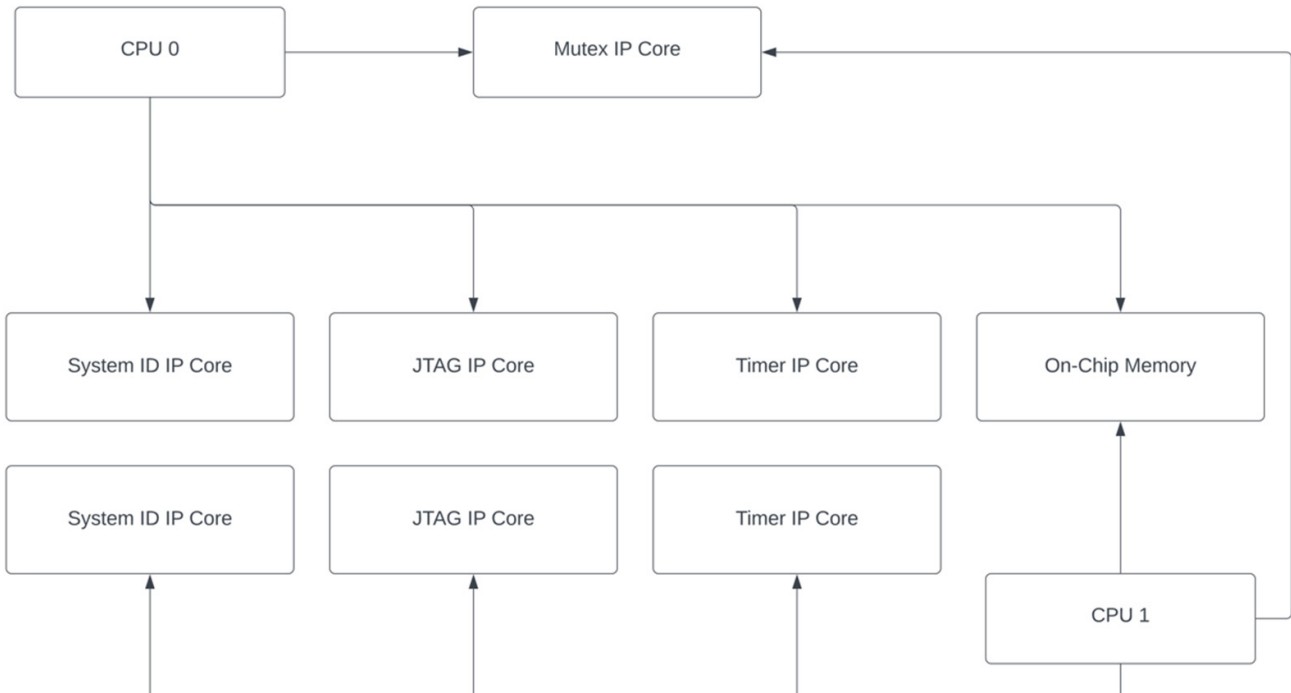

**Figure 11.** A diagram of a multi-NIOS system with peripherals.

### 3.2.2. The Structure of a Design in <.qsys> File

The information of an entire IP core-based design is stored in a <.qsys> file created by the platform designer software. By opening the file with a text editor, lines of information are stored in XML format. Figure 12 demonstrates an abstract diagram of the information in a <.qsys > file. Each block of the diagram is a part of the code. From the left to the right, the information is separated into six sections according to their categories. The name in each parentheses is the root name of the section. They are explained as follows:

- The first section is a list of all IP cores which are integrated in the system. It includes the position of each core that is displayed on Platform Designer and the base address of the IP cores;
- The second section stores the general information of the corresponding FPGA and the settings of the current design. For instance, the device name, device speed grade of the FPGA, the generating language (VHDL or verilog), etc.;
- The third section records all configurations of each IP core such as the NIOS II processor, timer, SDRAM controller, etc. Moreover, this section stores the settings adjusted by the developer on Platform Designer;
- The fourth section stores the connections of the Avalon interfaces. The connections among master and slave ports are listed;
- The fifth section stores the connections of the clock interfaces. The connections among the clock output and the clock input of each IP core are listed;
- The final section stores the connections of the reset interfaces. The connections among the reset output and the reset input of each IP core are listed.

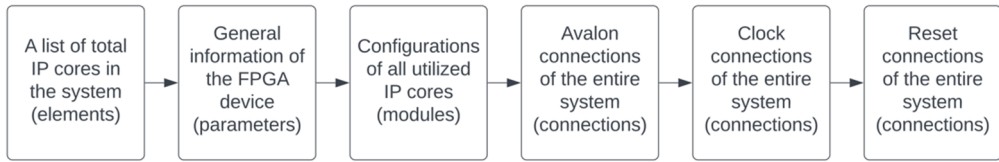

**Figure 12.** An abstract diagram of the information in a <.qsys > file.

### 3.2.3. The Structure of The Multi-processor Generator

The corresponding example runs on Terasic DE-4 (Stratix IV GX EP4SGX230C2). The Quartus version is 18.1. The multi-processor generator is a program based on python 3.9. By modifying the information in a <.qsys> file, a system can be directly customized without accomplishing the design on Platform Designer. The error-prone and time-consuming process of dot connections and configurations can be skipped. Figure 13 demonstrates a snapshot for the bus connections of a 10-core system. The clock and reset signals connect to all IP cores. The data and instruction master ports of all NIOS II CPUs need to be connected correctly to the slave port of the on-chip memory. To use the mutex core, the data master port of each NIOS II CPU is connected to the s1 port of the mutex core. A mistaken connection will cause malfunctional execution of a program. Figure 14 shows the abstract diagram of the multi-processor generator. The generator first asks developers about the file name and the number of NIOS II processors required in this system. Once the generator receives the input, it will calculate the memory space for each processor.

Figure 15 displays the mechanism of memory assignment. The total on-chip memory size is set to 1100 KB. Each processor will be assigned:

$$Memory\ space\ of\ each\ processor = \frac{1100 \times 1024}{n+1} \tag{1}$$

where $n$ is the number of processors.

The generator assigns the first section of the memory as a shared memory for user applications. CPU 0, CPU 1, . . . , CPU $n$ occupy a section of the rest of the memory. Then, the generator will assign addresses to each IP core and will store them in the elements section. Since the test runs on DE-4 board, the information of the Stratix IV GX EP4SGX230C2 will be written to the file. By using a loop function, $n$ numbers of NIOS II cores, timer, system ID, and JTAG core will be created. A single on-chip memory and a mutex core will also be created. The information is stored in module sections. Connections among IP cores will be generated afterward. Finally, the generator gives a customized solution of a multi-processor system by outputting a <.qsys> file. A DE-1 (Cyclone V 5CSEMA5F31C6) version of the multi-processor generator is also designed and tested.

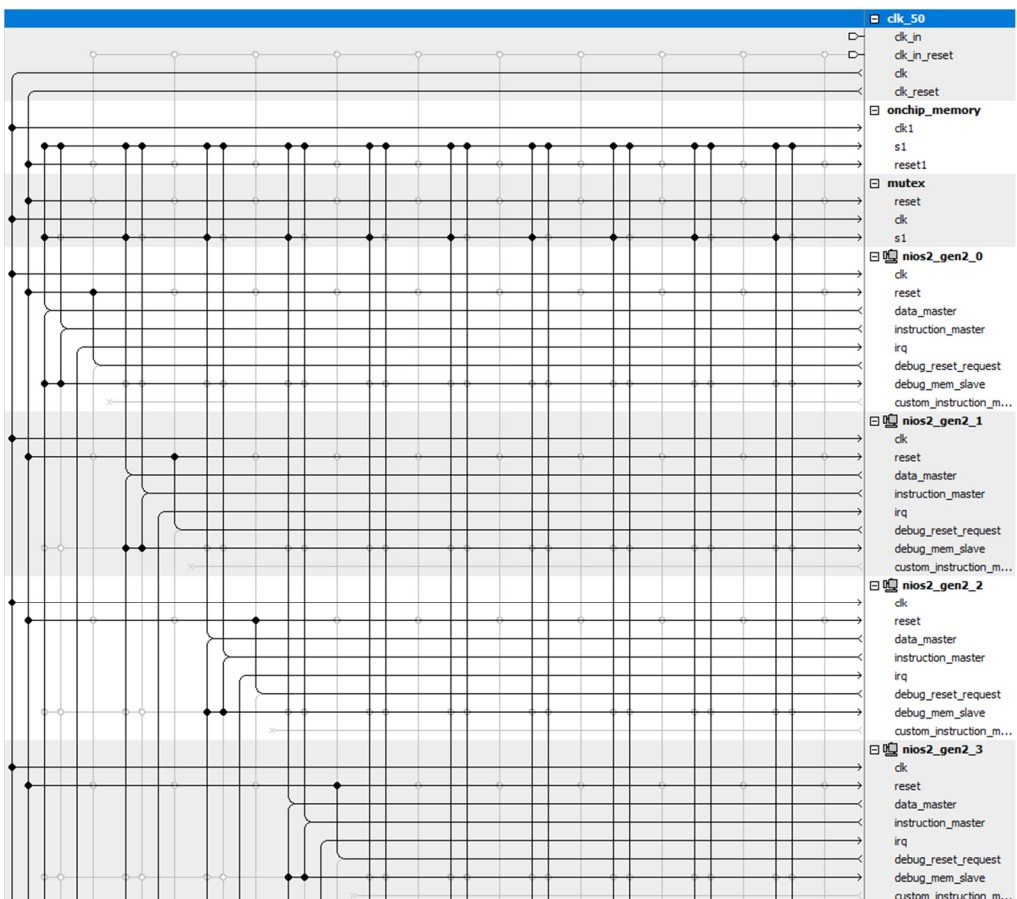

**Figure 13.** A snapshot for the bus connections of a 10-core system.

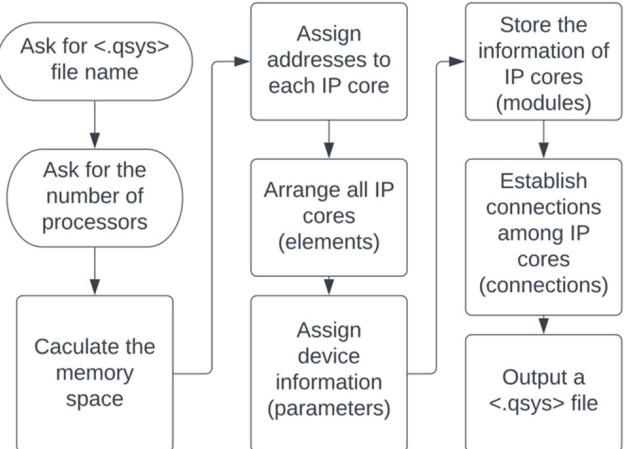

**Figure 14.** The abstract diagram of the multi-processor generator.

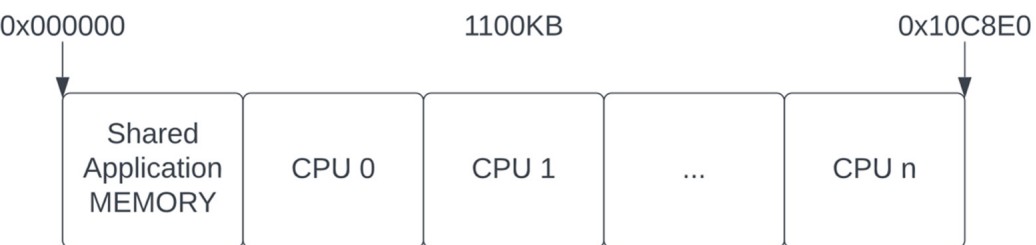

**Figure 15.** The mechanism of memory assignment.

### 3.3. Example Program on Multi-NIOS II Systems

3.3.1. A Parallel Application of FIR Filter

A parallel application of FIR Filter was implemented on a Terasic DE-4 (Stratix IV GX EP4SGX230C2) FPGA board. The Quartus version is 18.1. Figure 16 shows the pseudocodes of both host and slave NIOS II/f processors. This application is a discontinued FIR filter. A total of 256 elements of the inputs and system functions (128 elements each array) are stored in the codes. The application supports full-featured C library and HAL drivers.

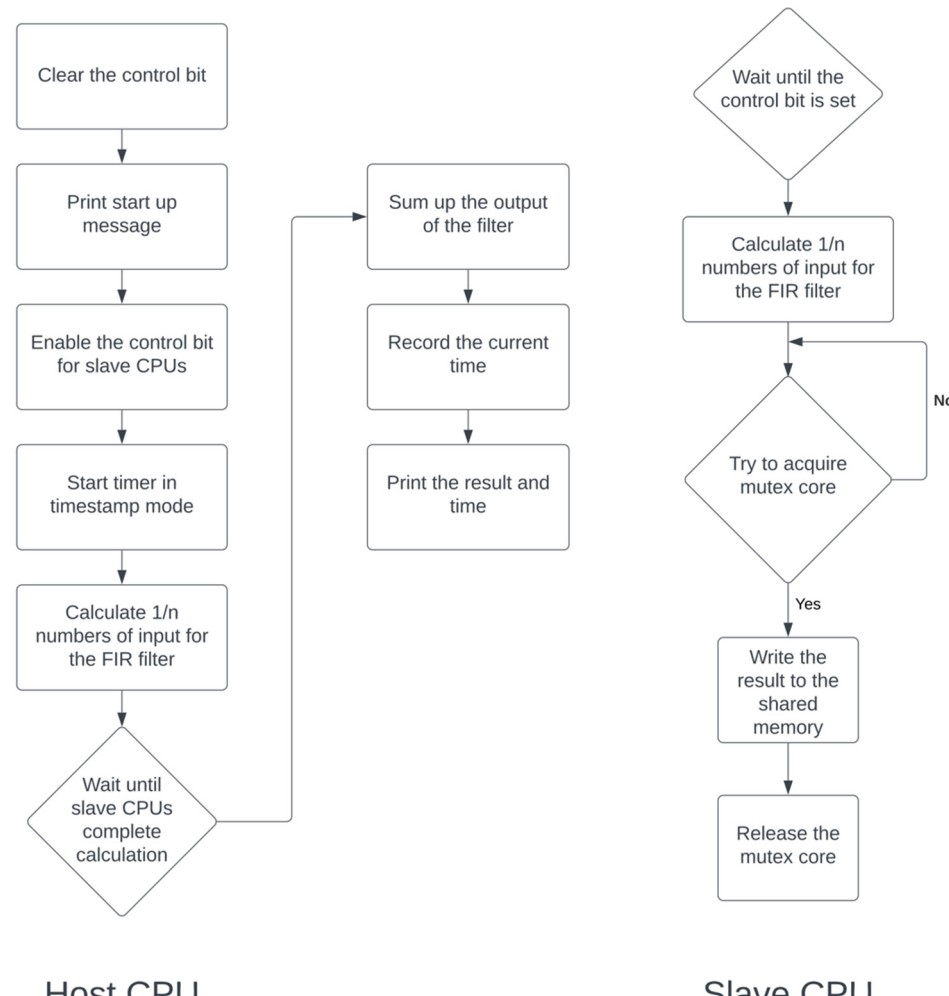

**Figure 16.** The mechanism of memory assignment.

When the application starts running, a host CPU will clear the control bit that was written by previous programs. Next, it prints a start message via JTAG terminal on the host PC. Then, the host CPU sets the control bit and turns on the timer in timestamp mode. The host CPU is not only responsible for controlling and reading signals from slave CPUs but also undertakes $128/n$ parts of the computational task. After the calculation, the host will wait for all slaves to finish their tasks. A complete signal written by slaves will be ensured by the host. Finally, the host will sum up the output of the filter, stop the timer, and print the result on the JTAG terminal.

Once the application starts running, slave CPUs will incessantly check the control bit. After the control bit is set, slaves compute the result of $128/n$ parts of the task. Then, slaves will try to acquire the ownership of the mutex core. One slave will hold the ownership till it finishes writing the result and a complete signal to the shared memory space. The host will calculate the sum of the result once it ensures all the complete signals for each slave are set.

### 3.3.2. DOS Commands Setup

To control all NIOS II CPUs straightforwardly, DOS commands are used to download FPGA images (.sof) and applications (.elf) for each CPU. Figure 17 demonstrates a pseudocode of the DOS setup. First, it specifies the Quartus directory on the host PC. After detecting the connection of the FGPA device, it will download the hardware and software design to the device. Developers can check the status of each CPU by monitoring each terminal window. Figure 18 displays a screenshot of terminals for running a 5-core system. The terminal on the top left is where the host CPU prints the result via JTAG core. The rest of the five windows are to download software applications to each CPU. It can be observed that CPUs execute the application on different memory spaces, which should be specified in the BSP generator of each CPU.

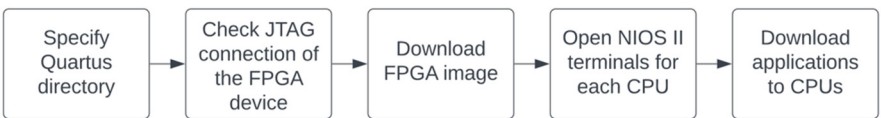

**Figure 17.** The pseudocode of the DOS setup.

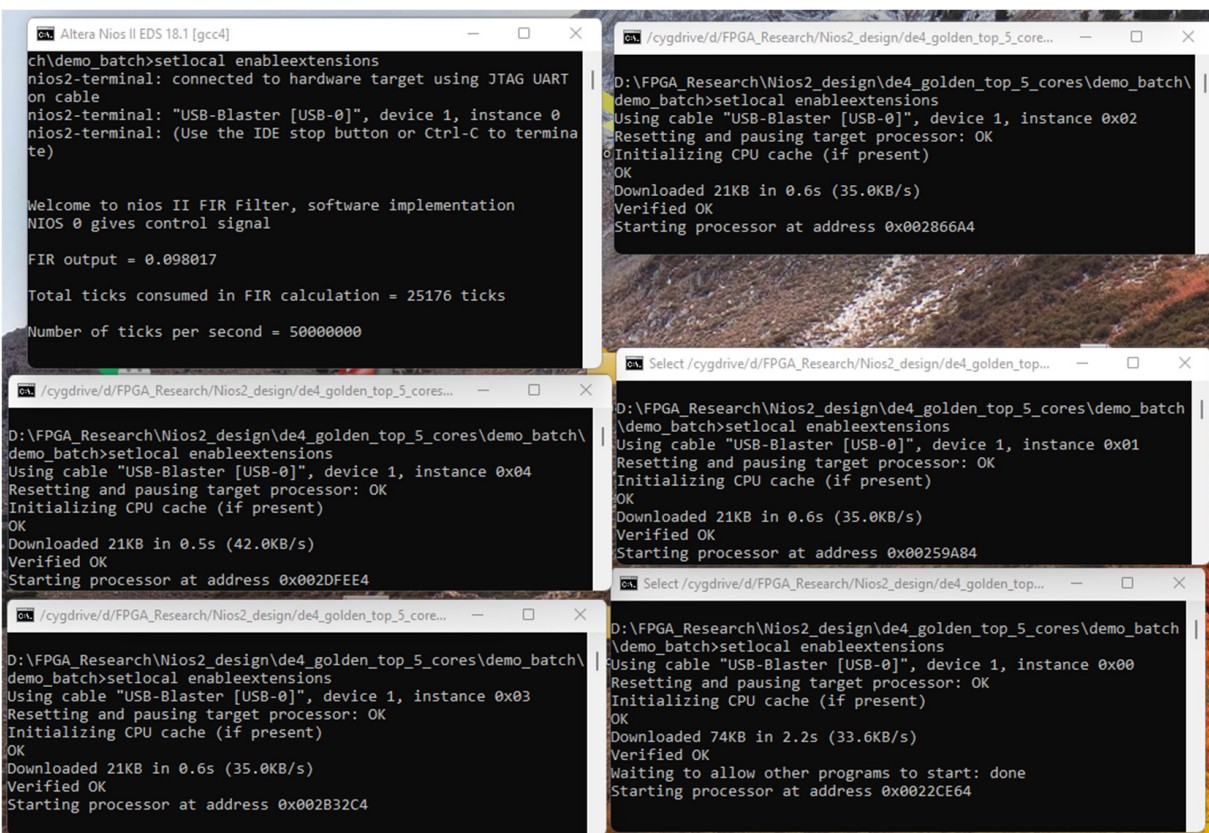

**Figure 18.** Terminals for a 5-core system.

## 4. Results

The application is tested on 1-, 2-, 5-, and 10-core NIOS II systems. The result is shown in Figure 19. From the top to the bottom, four screenshots of JTAG terminals for 1-, 2-, 5-, and 10-core systems are displayed. Each terminal is connected to the corresponding host CPU. The host CPU first displays a "Welcome" message once the host program starts executing. Then, it prints "NIOS 0 gives control signal" after it sets the control bit on the shared memory space. A timer starts counting. Meanwhile, the non-stop check on all slave CPUs detects the control bit has been set to "1". Slaves start to calculate the partial result of the FIR filter. Once a slave finishes the calculation, it tries to acquire the mutex and write

the result and finish signal on the shared memory. During the calculating period, the host CPU constantly checks the slaves' finish signal. The host reads the result from the shared memory after all the slaves' finish signals have been detected. The recording timer will then be read. In the end, all results will be printed by the host. The correct FIR outputs (0.098017) are calculated by four of them. Tick is the counting unit of the timer IP core. In the systems, its frequency is set as 50 Mhz. The total runtimes for four systems are 50,207, 35,350, 25,151, and 25,743 ticks. Approximately, a 2-core system is 29.5915% faster than a 1-core system. A 5-core system is 28.8515% faster than a 2-core system. A 10-core system is 2.3538% slower than a 5-core system.

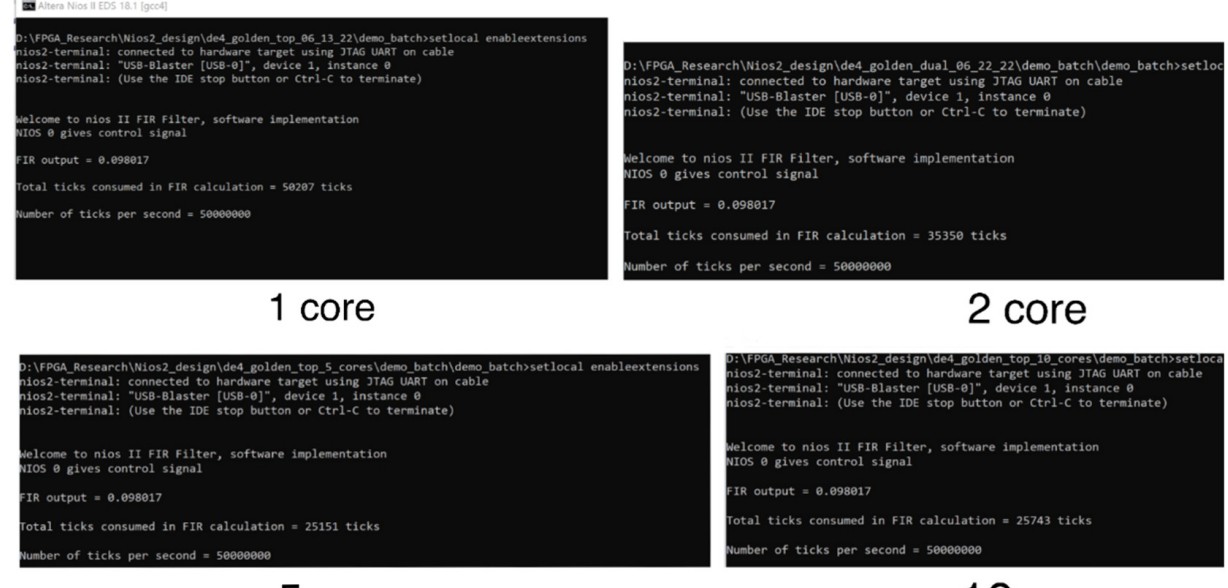

**Figure 19.** Screenshots for 1-, 2-, 5-, and 10-core systems from the top to the bottom.

In general, the use of soft-core processors can be divided into three categories:

1.  Microcontroller: In this case, soft-core processors are served as co-processors for a hard-core processor (such as Arm, Intel x86, or another soft-core processor). They are running programs (such as C codes and assembly codes) directly without any operating system on it for assisting the hard-core processor.

2.  Real-time processor: Unlike microcontrollers, a real-time OS (RTOS) is running on the soft-core processor. An RTOS manages system resources as a general-purpose OS and also guarantees the timing predictability of its applications. Without any RTOS modification, an RTOS can only run on one soft-core processor. However, other soft-core processors can serve as co-processors. Co-processors communicate with RTOSs and can be set as pre-processing units to reduce the overhead of the RTOS application so as to meet applications' strict timing constraint.

3.  Application processor: A soft-core processor running Linux. In a multi-processor system, other processors can be utilized as co-processors and communicate with the one core running Linux. The advantages are similar as they are in a real-time processor.

There are plenty of possible structures of a multi-NIOS II system setup. We chose one possible structure (NIOS II CPUs, UART unit, timer, OnChip RAM, system id, and mutex) and purpose a system generator to quickly set up a full-function multi-NIOS II system. The proposed system generator is capable of generating a multi-core microcontroller without any modification. Developers can directly test and run their program on it. For the use of a real-time processor and application processor, for instance, the size of FreeRTOS is in the range of 4000 to 9000 bytes. The size of a Linux kernel for NIOS II is around 3,000,000 bytes (3 MB). The size of the Stratix IV's on-chip memory we used in the design

is 1,100,000 bytes (1.1 MB). Therefore, there is enough memory space to install an RTOS on the multi-processor system generated by the generator. To use the generated NIOS II processor as an application processor, developers need to add additional RAM and ROM to install and boot Linux OS. In each case, the purposed system generator has greatly reduced the design time of a multi-core system and increased the efficiency of a design process. The testing result of the FIR filter has validated the practicality of this system generator.

## 5. Discussions

To reduce the inaccuracy of the runtime, the programs of host CPUs are implemented for a 1-core system in this example. Table 6 shows the task assignments for each system. The 1-core system calculates the entire input array of the FIR filter. The 2-core system split the array to half. A total of 26 elements of the array are calculated by one of the 3 cores, and 25 elements are calculated by one of the two cores in the 5-core system. In total, 11 elements are computed by a single core and 13 elements are computed by 9 cores in the 10-core system.

**Table 6.** Task assignments for each system.

|  | **1-Core** | **2-Core** | **5-Core** | **10-Core** |
|---|---|---|---|---|
| Executing elements per core | 128 by 1 core | 64 by 2 cores | 26 by 3 cores<br>25 by 2 cores | 13 by 9 cores<br>11 by 1 core |

Figure 20 displays a diagram of the runtime of the application vs. number of NIOS II cores. It can be observed that the FIR filter application is accelerated from a 5-core to 1-core system. However, the speed reduced when running on a 10-core system. The reason is that the size of the input array is only 128. The acceleration of the ten separate executions is offset by the frequent access of the mutex core. Compared with the 5-core system, five more slave CPUs need to gain the control of the mutex to read and write the shared memory. It can be deduced that a larger size (over 1000) of the input array will demonstrate the performance of a 10-core system. Moreover, a large input array will increase the portion of the computing task in the system. Ideally, it will boost the performances of the 2- and 5-core systems. During the counting time, the host CPU needs to communicate with the host PC via JTAG module, which causes the counting time to not be the exact same for each execution in one system. However, the counting error can be ignored since the difference of the value is less than 1%.

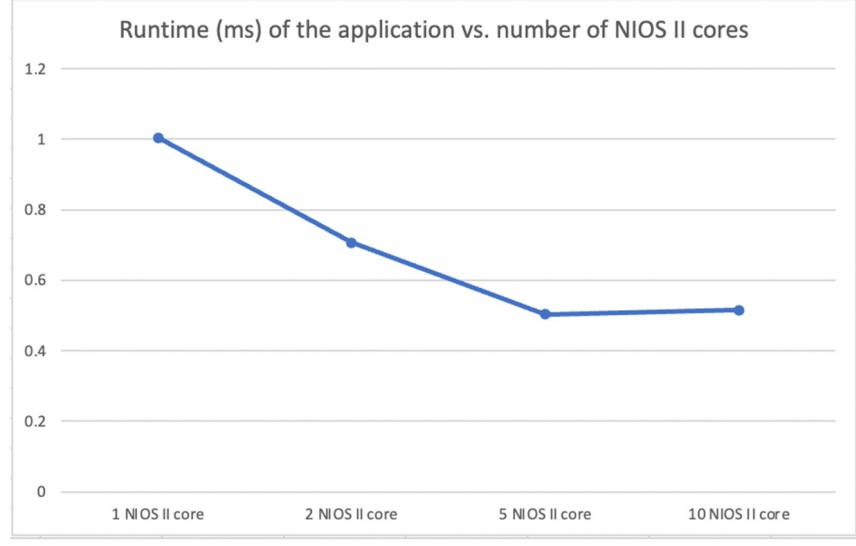

**Figure 20.** A diagram of the runtime of the application vs. number of NIOS II cores.

Table 7 shows the usage of logic resources of each system on a DE-4 board. The total logic utilization increases by adding an extra NIOS II core into the system. For a 10-core system, 15% of the logic resources are used, which indicates that around 60 NIOS II cores with their peripherals can be integrated on a DE-4 board. The gradually increased number of total registers conforms the percentage of the logic utilization. The generator assigns 1100 KB of on-chip memory to all four systems. However, total block memory bits increase as well. The reason is that the generator enables the data cache in each NIOS II/f core. The caches are constructed by an on-chip memory block. The Fmax value indicates that a larger system has lower performance compared with a smaller system.

**Table 7.** The usage of logic resources of each system on DE-4 board.

|  | Logic Utilization (Percentage) | Total Registers | Block Memory Bits | Fmax (Slow 900 mv 85C) |
|---|---|---|---|---|
| 1 core | 2% | 2051 | 8,864,512 (61%) | 199.92 MHz |
| 2 cores | 3% | 3923 | 8,928,896 (61%) | 144.38 MHz |
| 5 cores | 8% | 9724 | 9,124,288 (62%) | 141.48 MHz |
| 10 cores | 15% | 19,193 | 9,447,808 (65%) | 93.86 MHz |

## 6. Conclusions and Future Works

We found the hidden mechanism of the construction of hardware systems that has been utilized for Intel/Altera FPGAs. There are many structures for a multi-NIOS II system. We chose a structure that can be widely adopted. It includes critical components for a multi-NIOS II system (NIOS II processors, timers, JTAG UART units, shared memory, etc.). Based on the mechanism and the chosen components, we propose a novel multi-NIOS II system generator which can effortlessly build a practical hardware design that runs on Altera/Intel FPGAs. The necessary components and memory allocation will be automatically set up. According to different applied areas of a multi-NIOS II system (microcontroller, real-time processor, and application processor), the generated system can be further modified by developers. In each case, the purposed system generator has greatly reduced the design time of a multi-core system and increased the efficiency of a design process. The background of the HW/SW co-design on Altera/Intel FPGAs was described. The NIOS II soft processor core and related IP peripherals was introduced. The structure of a <.qsys> system was explained. Moreover, we present the mechanism of the system generator. In the end, a parallel application of the FIR filter was implemented on 1-, 2-, 5-, and 10-core systems. Approximately, a 2-core system is 29.5915% faster than a 1-core system. A 5-core system is 28.8515% faster than a 2-core system. A 10-core system is 2.3538% slower than a 5-core system. The result was discussed. The testing result of the FIR filter validated the practicality of this system generator.

The boost performance of a 10-core system is based on assumption. A larger input array will be adopted in the future. Moreover, we will add an ARM processor into the generator. The communication between the Linux ARM and multi-NIOS II system will be established. The ARM processor will be the host CPU and NIOS II CPUs will serve as co-processors. We will then use the generated system to discuss the acceleration of algorithms in the image processing area. The example program and the multi-processor generator can be downloaded at: https://drive.google.com/file/d/1BFPrwgh7YKfFFozlRiTVOIe6kqygC0WV/view?usp=sharing (accessed on 5 September 2022).

**Author Contributions:** Conceptualization, H.C.; methodology, H.C.; software, H.C.; validation, U.M.-B.; writing-original draft preparation, H.C.; writing-reviewing and editing, U.M.-B.; visualization, H.C. and U.M.-B.; supervision, U.M.-B.; project administration, U.M.-B.; funding acquisition, U.M.-B. All authors have read and agreed to the published version of the manuscript.

**Funding:** This research received no external funding.

**Data Availability Statement:** https://drive.google.com/file/d/1BFPrwgh7YKfFFozlRiTVOIe6kqygC0 WV/view?usp=sharing (accessed on 5 September 2022).

**Conflicts of Interest:** The authors declare no conflict of interest.

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
