# Peer review of "XML-Based Automatic NIOS II Multi-Processor System Generation for Intel FPGAs"

_electronics, doi:10.3390/electronics11182840_

Round 1

Reviewer 1 Report

The paper is outstanding, there are some minors mistakes, please correct it, please see the attached file

Author Response

Thank you for kindly reviewing our paper. The answers of corresponding comments has been attached in the file

Reviewer 2 Report

The paper tackles a relevant topic related to rapid prototyping of hardware/software co-design based on FPGAs, proposing a novel system generator to effortlessly design a multiple NIOS II soft-processor core systems.

The FPGA overview is well presented. In the methodology section the authors present the HW/SW co-design process, the mechanism of multiprocessor generator and the programming on Multi-NIOS II systems. How is the structure of Xilinx FPGAs compared to those from Intel-Altera?

The section 4, results, is too brief and do not present sufficient the project results and contributions. Figure 18 must be clearer and discussed more extensively. The paper is relatively full, however it needs some improvements:

- The improvements achieved with the presented project should have been more clearly highlighted.

- How would the complexity of the Multi-NIOS II systems be reflected in the real-time system responses?

- Can a comparison be made with MicroBlaze -Xilinx?

Besides discussions section, the task assignments and the runtime of the application vs. number of NIOS II cores are well described. What operating system was used with NIOS II?  Maybe the related work section should be written separately. The reference section is relatively OLD (articles from the period 2020-2022 are not cited), it does NOT cite new and relevant articles in the research area.

In particular, it is not clear whether the authors' contributions and research innovation to this publication.

Author Response

Thank you for your patiently review. The answers of corresponding comments has been attached in the file

Round 2

Reviewer 2 Report

The paper has been improved sufficiently to be accepted for publication.